

# Glutathione-mediated changes in productivity, photosynthetic efficiency, osmolytes, and antioxidant capacity of common beans (*Phaseolus vulgaris*) grown under water deficit

Taia A. Abd El Mageed[1], Wael Semida[2], Khoulood A. Hemida[3], Mohammed A.H. Gyushi[2], Mostafa M. Rady[4], Abdelsattar Abdelkhalik[2], Othmane Merah[5,6], Marian Brestic[7,8], Heba I. Mohamed[9], Ayman El Sabagh[10,11] and Magdi T. Abdelhamid[11,12]

[1] Soil and Water Department, Faculty of Agriculture, Fayoum University, Fayoum, Egypt
[2] Horticulture Department, Faculty of Agriculture, Fayoum University, Fayoum, Egypt
[3] Botany Department, Faculty of Science, Fayoum University, Fayoum, Egypt
[4] Botany Department, Faculty of Agriculture, Fayoum University, Fayoum, Egypt
[5] Laboratoire de Chimie Agro-industrielle, Université de Toulouse, Toulouse, Toulouse, France
[6] IUT A, Département Génie Biologique, Université Paul Sabatier-Toulouse III, Auch, France
[7] Plant Physiology, Slovak University of Agriculture, Nitra, Nitra, Slovakia
[8] Institute of Plant and Environmental Sciences, Slovak University of Agriculture in Nitra, A. Hlinku 2, Nitra, Slovakia
[9] Biological and Geological Sciences Department, Faculty of Education, Ain Shams University, Cairo, Egypt
[10] Department of Agronomy, Faculty of Agriculture, Kafrelsheikh University, Kafr Al-Sheik, Egypt
[11] Botany Department, National Research Centre, Cairo, Egypt
[12] Department of Soil and Crop Sciences, Texas A&M University, College Station, TX, United States of America

Corresponding author
Magdi T. Abdelhamid,
magdi.abdelhamid@yahoo.com

## ABSTRACT

Globally, salinity and drought are severe abiotic stresses that presently threaten vegetable production. This study investigates the potential exogenously-applied glutathione (GSH) to relieve water deficits on *Phaseolus vulgaris* plants cultivated in saline soil conditions (6.22 dS m$^{-1}$) by evaluating agronomic, stability index of membrane, water satatus, osmolytes, and antioxidant capacity responses. During two open field growing seasons (2017 and 2018), foliar spraying of glutathione (GSH) at 0.5 (GSH$_1$) or 1.0 (GSH$_1$) mM and three irrigation rates (I$_{100}$ = 100%, I$_{80}$ = 80% and I$_{60}$ = 60% of the crop evapotranspiration) were applied to common bean plants. Water deficits significantly decreased common bean growth, green pods yield, integrity of the membranes, plant water status, SPAD chlorophyll index, and photosynthetic capacity (F$_v$/F$_m$, PI), while not improving the irrigation use efficiency (IUE) compared to full irrigation. Foliar-applied GSH markedly lessened drought-induced damages to bean plants, by enhancing the above variables. The integrative I$_{80}$ + GSH$_1$ or GSH$_2$ and I$_{60}$ + GSH$_1$ or GSH$_2$ elevated the IUE and exceeded the full irrigation without GSH application (I$_{100}$) treatment by 38% and 37%, and 33% and 28%, respectively. Drought stress increased proline and total soluble sugars content while decreased the total free amino acids content. However, GSH-supplemented drought-stressed plants mediated

further increases in all analyzed osmolytes contents. Exogenous GSH enhanced the common bean antioxidative machinery, being promoted the glutathione and ascorbic acid content as well as up-regulated the activity of superoxide dismutase, catalase, ascorbate peroxidase, and glutathione peroxidase. These findings demonstrate the efficacy of exogenous GSH in alleviating water deficit in bean plants cultivated in salty soil.

## INTRODUCTION

Plants are subjected to recurring abiotic stresses during their growth, threatening the production of vegetable crops. Water stress and salinity alone or in combination are the main yield-limiting factors (*Awad et al., 2012*; *Semida et al., 2020*). Forecasts based on the integration of crop growth models and climate change estimated further yield losses (*Waqas et al., 2019*), associated with population growth that would require an increase in food production, consequently increasing the demands for irrigated agriculture, which requires heightening the irrigation use efficiency (IUE) (*Abdelkhalik et al., 2019c*; *Abd El-mageed et al., 2021*).

Soil water deficit alters many vital processes and causes severe damage to the plant. A plant's primary response to soil water reduction is a decrease in cell turgor pressure, which causes osmotic stress, resulting in several cellular signaling pathways that disrupt different physio-biochemical activities in plant cells (*Xiong & Zhu, 2002*; *Abdelkhalik et al., 2019c*). Later, drought/osmotic stress leads to excessive accumulation of reactive oxygen species (ROS; *e.g.*, OH $-$, $H_2O_2$, and $O2^{\bullet -}$) produced from different plant organelles; mitochondria, chloroplasts, and peroxisomes (*Rady et al., 2021*; *Abd El-mageed et al., 2022*). Hyperproduction of ROS destructs the normal balance between ROS formation and scavenging that not only inhibits several enzymes activity but also triggers cellular oxidative damage like protein, DNA, and lipids even can lead to cell damage (*Zhu et al., 2020*; *Abd El-mageed et al., 2022*). Simultaneously, ROS provokes chlorophyll degradation and membrane lipid peroxidation, decreasing membrane stability, selectivity, and fluidity, and disturbing cell redox homeostasis (*Rady, Taha & Kusvuran, 2018*; *Semida et al., 2021a*). Under water stress, the lower tissue water content becomes limiting for physiological and biochemical processes in the plant (*Tombesi et al., 2015*; *Abdelkhalik et al., 2019a*). Besides stomatal closure is a primary factor that decreases photosynthesis, ROS generated in the chloroplasts can damage the photosynthetic pigments, thylakoid membrane, and enzymes, as well as decrease or inhibits the photosynthetic capacity of photosystem II (PSII) (*Ma, Dias & Freitas, 2020*; *Semida et al., 2021b*).

The plant has adapted several regulatory signaling mechanisms to withstand water stress, including activating the plant's defense system, which helps the plant modulate metabolism, maintain protective regimes, and redox homeostasis (*Rady et al., 2019*). The abundance of osmoprotectants in plant tissues involves osmotic adjustment, the maintenance of cell

turgor, and the control of water influx/efflux (*Blum, 2017*; *Turner, 2018*). Furthermore, increasing the activity of antioxidants both enzymatic and non-enzymatic helps to reduce ROS and protect membrane lipids from peroxidation under drought stress (*Farooq et al., 2009*; *Semida, Hemida & Rady, 2018*; *Zhu et al., 2020*). However, a continuous water deficit can trigger an imbalance among scavenge and the production of ROS, which inhibits the antioxidative machinery's scavenging action. In that case, the endogenous plant defense mechanisms are unable to completely protect plants from the harmful effects of drought stress, thus impairing plant growth and reducing yield (*Pal et al., 2016*; *Rady et al., 2021*). Therefore, exogenous use of auxiliary compounds such as antioxidative compounds or another effective approach to support plant defense system, allows plants to perform well under water deficit.

Glutathione (GSH) is a major free thiol tripeptide with a low molecular weight that acts as an antioxidant to relieve environmental stresses in several ways (*Ding et al., 2016*; *Ashraf et al., 2019*). GSH participates in the antioxidant defense, hormone or redox molecule signaling, transmembrane of amino acids transport, and detoxification of ROS, methylglyoxal, and xenobiotics (*Hasanuzzaman, Nahar & Anee, 2017*; *Zhou et al., 2019*). GSH is a small molecular weight of a thiol group that presents the GSH as a powerful ROS scavenger (*Gill et al., 2017*), it can also scavenge ROS indirectly through the AsA-GSH cycle, which removes harmful peroxides (*Rady & Hemida, 2016*; *Hossain et al., 2017*). GSH is an active redox compound that is typically found in reduced glutathione (GSH) form which may be oxidized by ROS to disulfide glutathione (GSSG) form that is recirculated to GSH by NADPH-dependent glutathione reductase (*Noctor et al., 2012*). The balance of GSH and GSSG in the cell is related to its redox state, in the sense that a higher GSH or GSH/GSSG ratio is important for many physiological mechanisms and regulates various adaptations to abiotic stress resilience (*Ding et al., 2016*; *Hossain et al., 2017*). Furthermore, GSH contributes to plasma membrane stability by reducing lipid peroxidation, as well as osmotic adjustment for abiotic stress tolerance (*Hasanuzzaman, Nahar & Anee, 2017*; *Rehman et al., 2021*). Exogenous GSH has been shown in recent studies to improve plant growth, leaf water status, and photosynthesis, and reduce oxidative damage indicators, along with up-regulated some osmolytes, the antioxidant capacity, and induced cellular redox hemostasis under abiotic stress such as drought stress (*Nahar et al., 2015*), high temperature (*Ding et al., 2016*) and salt stress (*Rady & Hemida, 2016*).

The *Phaseolus vulgaris*, a common bean, is among the world's largest legume crops, which are produced and consumed on a large scale (*Biddle, 2017*). The world's green pod production is approximately 27 million Mg, grown on an area of approximately 1.65 million ha, and Egypt is one of the world's major producers and exporters of green beans (*FAOSTAT, 2020*). Irrigation is critical at all growth stages of bean plants, which requires adequate water to achieve an important yield (*Sezen et al., 2008*). Furthermore, common beans are considered a highly salt-sensitive crop, being exhibit growth reduction and injury symptoms upon exposure to salinity (*Maas & Grattan, 1999*).

However, little information exists regarding the influence of the foliar application of antioxidative compounds like GSH on drought-stressed common bean plants cultivated in salt-affected soil in open field conditions. Therefore, the current research was intended

to look into the potential use of GSH to reduce the adverse consequences of water shortage on common beans. In this study, the potential changes in physio-biochemical attributes, osmolytes, and antioxidative molecules of *Phaseolus vulgaris* were examined under combined exogenously-applied GSH and water deficit. Furthermore, the expected enhancements in water status, membrane stability, photosynthetic efficiency, growth parameters, pod yield, and IUE of common bean induced by GSH under water deficits were assessed.

## MATERIAL AND METHODS

### Experimental field site

The field experiments were performed on a private farm in Fayoum governorate, Egypt, 29.5004 N, 30.8767 E, during the two seasons of 2017 and 2018. Before the start of each experiment, soil samples were collected to a depth of 25 cm from the experimental site and analyzed for physical and chemical properties (Table 1) according to *Klute (1986)* and *Page, Miller & Keeney (1982)*. The average monthly climatic data of El-Fayoum during the study period (September −November) are presented in Table 2. The study area has a hyperarid climate as identified by the aridity index (*Ponce, Pandey & Ercan, 2000*).

### Irrigation applied (IA) and treatments

Every two days, the growing *Phaseolus vulgaris* plants were irrigated with varying amounts of irrigation water. The amount of irrigation applied (m$^3$) to each experimental unit was calculated by using the following equation:

$$\text{Irrigation applied} = \frac{A \times ETc \times Ii \times Kr}{Ea \times 1,000 \times (1 - LR)}$$

where; A is the experimental unit area (m$^2$), $ET_c$ is the crop evapotranspiration (mm day$^{-1}$), Ii is the intervals between irrigation events (day), Kr is the covering factor, Ea is the application efficiency (%), and LR is the leaching requirements.

The daily reference evapotranspiration; $ET_o$ (mm day$^{-1}$) was computed using the evaporation registered from class A pan ($E_{pan}$) and appropriate pan coefficient ($K_{pan}$) for the experimental area as follows:

$$ET_o = E_{pan} \times K_{pan}.$$

The crop water evapotranspiration ($ET_C$) was estimated using the crop coefficient according to the following equation:

$$ET_c = ET_o \times K_c$$

where $ET_o$ = the reference evapotranspiration (mm day$^{-1}$) and $K_c$ = the crop coefficient. The duration of the initial, crop development, mid-season, and late-season stages were 15, 25, 25, and 10 days, respectively. The *Phaseolus vulgaris* $K_c$ according to *Allen et al. (1998)* was 0.50, 1.05, and 0.90, corresponding to the initial, mid, and end stages, respectively.

In a preliminary pot experiment the GSH was foliar-applied with 0.5 and 1.0 mM at 25 days after sowing and applied again after 10 days. The spray solution was prepared
**Table 1  Some initial physico-chemical characteristics of the studied soil.**

| EC (dS/m) | pH | OM % | CaCO$_3$ | Particle size distribution | | | Texture class | $\rho$d g.cm$^{-3}$ | K$_{sat}$ cm h$^{-1}$ | Soil moisture content at | | |
|---|---|---|---|---|---|---|---|---|---|---|---|---|
| | | | | Sand % | Silt % | Clay % | | | | FC % | WP % | AW % |
| 6.22 | 7.66 | 1.13 | 4.51 | 13.0 | 12.8 | 74.2 | LS | 1.58 | 2.10 | 21.03 | 10.55 | 10.48 |

Notes.
EC, electrical conductivity; OM, Organic matter content %; LS, loamy sand; $\rho$d, Bulk density; K$_{sat}$, Hydraulic conductivity; FC, Field capacity; WP, wilting point; AW, Available water.

**Table 2  Monthly weather data at Fayoum, Egypt as an average for 2017–2018.**

| Month | T$_{max}$ (°C) | $^a$T$_{min}$ (°C) | T$_{avg}$ (°C) | RH$_{avg}$ (%) | U$_2$ ms$^{-1}$ | ETo (mmd$^{-1}$) |
|---|---|---|---|---|---|---|
| September | 38.3 | 23.6 | 30.95 | 37.0 | 2.1 | 5.85 |
| October | 34.0 | 22.4 | 28.2 | 40.0 | 1.95 | 4.7 |
| November | 27.8 | 15.4 | 21.6 | 41.5 | 2.2 | 2.15 |

Notes.
T$_{max}$, T$_{avg}$, and T$_{min}$ are average, maximum, and minimum temperatures, respectively.
RH$_{avg}$, average relative humidity; U$_2$, average wind speed; E$_P$, average of measured pan evaporation class A.

by dissolving GSH in distilled water with adding Tween-20 (0.1%, v/v) as a surfactant to increase its retention on the leaves, thus fast penetration through the leaves. For each experimental plot (9 m$^2$), a volume of 9 L of spraying solution was specified per each application time. Foliar spraying GSH at the concentrations and times obtained from the initial pot experiment achieved the highest growth of green bean plants grown under three irrigation rates 100, 80, and 60% of ET$_c$. For the current study, there were two factors including GSH (0, 0.5 mM = GSH$_1$ and 1.0 mM = GSH$_2$), and irrigation regimes as a percentage of the ET$_c$ (100% = I$_{100}$, 80% = I$_{80}$, and 60% = I$_{60}$). Therefore, nine treatments were established as follows; I$_{100}$ (full irrigation without GSH application), I$_{80}$ (irrigation with 80% of the ET$_c$ without GSH application), I$_{60}$ (irrigation with 60% of the ET$_c$ without GSH application), I$_{100}$ + GSH$_1$, I$_{100}$ + GSH$_2$, I$_{80}$ + GSH$_1$, I$_{80}$ + GSH$_2$, I$_{60}$ + GSH$_1$, and I$_{60}$ + GSH$_2$.

## Plant materials and experimental layout

The experimental layout was a randomized complete block design with three replications. The total experimental included 27 plots; each one was approximately 9 m$^2$ (15 m in length × 0.6 m row width) each plot included 2 planting rows placed 30 cm apart with a distance of 10 cm between plants within rows. Two drip lines were placed 0.3 m apart in each elementary test plot. *Phaseolus vulgaris* (cv. Bronco) seeds were planted on 6 September and harvested on 28 November in both growing seasons. All treatments were separated and surrounded by a 1m non-irrigated area. Plants were adequately watered during the first irrigation. One week after complete germination, irrigation treatments were started. All experimental units received identical doses of N, P$_2$O$_5$, and K150 kg N ha$^{-1}$, 60 kg P ha$^{-1}$, and 70 K kg ha$^{-1}$ orderly. The other cultural practices for commercial bean production were carried out according to the instructions of the Ministry of Agriculture and Land Reclamation.

## Growth and yield-related attributes

After 50 days of sowing in each season, three random plants from each experimental unit were taken to measure morphological characteristics, and another group of three plants was taken to determine physio-biochemical traits. The lengths of the shoots were measured on a meter scale, and the number of leaves per plant was counted. A digital planimeter (Planix 7, Tamaya Technics Inc., Tokyo, Japan) was used to measure the leaf area per plant. Plant shoots were weighed to determine their fresh weight before being oven dried at 70 °C until their weight stabilized. Green pods were collected at the harvest stage from all plants in each plot to determine the average number of pods per plant, green pod weight per plant, and green pod yield per hectare in a ton.

## Leaf relative water content, membrane stability, and irrigation use efficiency

The relative water content (%) in bean leaves were assessed (*Osman & Rady, 2014*). After excluding the midrib, 2-cm discs were taken and the fresh mass (FM) was weighed. Immediately, the discs were submerged in distilled water for 24 h, after which they were extracted and weighed to determine the saturated mass (SM). The dry mass of discs after dehydrating at 70 °C for 48 h was recorded. The RWC was calculated using the formula:

$$RWC(\%) = (FM - DM)/(SM - DM) \times 100.$$

Duplicate samples of fully-expanded fresh leaves tissue, each weighing 0.2 g were used to determine the stability index of the cellular membrane (MSI) (*Abdelkhalik et al., 2019b*). The first leaf sample was placed in a test tube with 10 ml of double-distilled water and heated in a water bath at 40 °C for 30 min. The electrical conductivity of the solution was measured and denoted a C1. The same previous steps were performed with the second leaf sample but the samples were heated at 100 °C for 10 min, and electric conductivity was measured and denoted a C2. The MSI was calculated as follows:

$$MSI(\%) = 1 - (C1/C2) \times 100.$$

Irrigation use efficiency (IUE) values were calculated for different treatments as kg green pods per cubic meter (m3) of applied water using the following equation (*Jensen, 1983*):

$$IUE = Pods\ yield(kg\ ha^{-1})/water\ applied(m^3ha^{-1}).$$

## Photosynthetic efficiency

Using a chlorophyll meter; SPAD-502 (Minolta, Tokyo, Japan) fully developed leaves were collected from the top of each plant to determine the relative chlorophyll content (SPAD value). The chlorophyll fluorescence parameters were measured with one leaf per plant on two different sunny days using a portable fluorometer (Handy PEA, Hansatech Instruments Ltd., Kings Lynn, UK). The maximum quantum yield (Fv/Fm) was calculated using the formula: $F_v/F_m = (F_m - F_0)/F_m$ (*Maxwell & Johnson, 2000*). According to *Clark et al. (2000)*, the photosynthetic performance index (PI) was determined.

## Quantification of osmoprotectants and non-enzymatic antioxidant contents

Free proline content was quantified using the rapid colorimetric method described by *Bates, Waldren & Teare (1973)*. *Rosen*'s (*1957*) method was used to determine the total free amino acid content of dry leaves. Leaf-soluble sugar content was determined after extraction with 96% (v/v) ethanol, as outlined by *Irigoyen, Einerich & Sánchez-Díaz (1992)*. The extract was reacted with anthrone reagent, the obtained mix was boiled for 10 min. After cooling, the samples were read at 625 nm using a spectrophotometer (a Bausch and Lomb-2000).

Leaf ascorbic acid (AsA) content was extracted and quantified according to the methods of *Kampfenkel, Van Montagu & Inzé (1995)*. To assay AsA content, a leaf sample of 1.0 g was homogenized and extracted with 5% (w/v) trichloroacetic acid (TCA) with liquid $N_2$ then the mixture was centrifuged (15,600 $\times$ g, 4 °C, 5 min). 1.0 ml of the supernatant was carefully taken and placed in the vessel tube with 0.5% (v/v) nethylmaleimide, 10 mM DTT, 10% (w/v) TCA, 4% (v/v) 2,2′-dipyridyl, 42% (v/v) $H_3PO_4$, 3% (w/v) $FeCl_3$, in addition to 0.2 M phosphate buffer with pH 7.4. For GSH contents determination by *Griffith (1980)*, homogenization of fresh leaf tissue (50 mg) was exercised in two mL of 2% (v/v) metaphosphoric acid, and centrifugation was then applied at 17,000 $\times$ g for 10 min. The supernatant was neutralized with sodium citrate of 10% (w/v). Assessments of 3 replicates were made for each sample. A composition of 700 µL of 0.3 mM NADPH, 100 µL of 6 mM 5,50 -dithiol-bis-2- nitro benzoic acid, 100 µL distilled water, and 100 µL of the extract was of each assay (1.0 mL) that was stabilized at 25 °C for 3–4 min, and GSH reductase (10 µL of 50 units $mL^{-1}$) was then added and the absorbances were read at 412 nm to calculate GSH contents from a standard curve.

## Enzymatic antioxidant assay

For obtaining the enzyme extracts, 200 mg freeze-fresh leaf was homogenized in a cold mortar with 2 ml of extraction buffer prepared from potassium phosphate buffer (100 mM, pH 7.0) containing 0.1 mM ethylenediaminetetraacetic acid (EDTA) (*Bradford, 1976*). For assaying the APX activity (µmol $H_2O_2$ $min^{-1}$ $g^{-1}$ protein) (*Nakano & Asada, 1981*), 2 mM AsA was added to the extraction buffer. The homogenate was filtered and then centrifuged at 12,000 $\times$ g for 15 min. All steps were completed under 4 °C. The mixture (2 ml) was spotted for 2 min at 290 nm with a spectrophotometer measuring the AsA oxidation, and an extinction coefficient of 2.8 $mM^{-1}$ $cm^{-1}$ was used. The activity (µmol $H_2O_2$ $min^{-1}$ $g^{-1}$ protein) of CAT (EC 1.11.1.6) was quantified as described by *Havir & McHale (1987)*, by measuring the decrease in absorbance read at 240 nm caused by $H_2O_2$ breakdown ($\varepsilon$ = 36 $M^{-1}$ $cm^{-1}$). The activity (U $mg^{-1}$ protein) of SOD (EC 1.15.1.1) was measured by determining its ability to inhibit nitro blue tetrazolium (NBT) photochemical decrement (*Beauchamp & Fridovich, 1971*). The enzyme amount required to inhibit half of the NBT photoreduction rate (%) was assigned as one unit of SOD activity. As outlined by *Martinez et al. (2018)* the GPX activity was quantified with a glutathione peroxidase assay kit (Ref. ab102530; Abcam, Cambridge, UK) measuring the reduction of NADPH at 340 nm, and extinction coefficient of 6.22 $mM^{-1}$ $cm^{-1}$ was used.

## Statistical analysis

Data from both field experimental seasons were analyzed using GenStat 19th Edition (VSN International Ltd, Hemel Hempstead, UK). Differences between the treatments were compared using Tukey's Honest Significant Difference test at $P \leq 0.05$.

## RESULTS

### Growth characteristics and green pods yield

Results in Table 3 exhibited that deficit irrigation ($I_{80}$ or $I_{60}$) unfavorable affected all growth parameters; *i.e.,* shoot length, plant leaf area, the number of leaves per plant, and shoot dry weight plant$^{-1}$. However, foliar-applied GSH (0.5 or 1.0 mM) corrected this growth inhibition and increased all growth parameters compared to deficit irrigation treatment ($I_{80}$, or $I_{60}$) in both seasons. Generally, the maximum values of all analyzed growth traits were recorded under $I_{100}$ + GSH$_1$ treatment. However, foliar-applied GSH (0.5 mM) to 20% water-stressed common bean plants increased the aforementioned growth parameters and registered similar or higher values than fully irrigation plants untreated with GSH ($I_{100}$ treatment).

In both seasons, the gradual reduction of irrigation from 100% ($I_{100}$) to 60% ($I_{60}$) of ETc significantly decreased gradually the number of pods (up to 61%), green pods weight (up to 48%), and green pods yield ha$^{-1}$ (up to 47%) (Table 4). However, exogenously-applied GSH to common bean plants grown under water stress recovered the yield losses by inducing considerable increases in the number of pods, green pods weight, and green pods yield compared to the respective control. Foliar spraying bean plants grown under 20% water deficit with GSH (0.5 mM) increased the abovementioned traits by 38% and 37%, 24% and 23%, and 24% and 18% (seasons average) respectively, when compared with the corresponding control, with similar values as those observed under full irrigation ($I_{100}$). However, integrative $I_{60}$ + GSH$_1$ or GSH$_2$ elevated the green pod's yield and its component compared to the corresponding control but did not reach those observed under optimum irrigation without GSH application ($I_{100}$).

### Membrane integrity, water status, and irrigation use efficiency

As presented in Table 5, deficit irrigation ($I_{80}$ and $I_{60}$) induced stress in bean plants, being reduced the membrane stability index (MSI) by 14% and 40% and leaf relative water contents (RWC) by 6% and 17% (seasons average) respectively, compared to well-watered plants without application of GSH ($I_{100}$). Nevertheless, GSH supplementation attenuated the water deficit-induced damages as the same values of MSI and RWC were observed under fully irrigated plants untreated with GSH ($I_{100}$). Reducing irrigation by up to 60% of ETc markedly decreased the irrigation use efficiency (IUE) by 13% in comparison with full irrigation ($I_{100}$). However, combined externally applied GSH and water deficit substantially elevated the IUE. The integrative $I_{80}$ + GSH$_1$ or GSH$_2$ and $I_{60}$ + GSH$_1$ or GSH$_2$ exceeded the full irrigation without GSH application ($I_{100}$) treatment by 38% and 37%, and 33% and 28% (seasons average) respectively.

**Table 3** Effect of exogenous spray applications of glutathione (GSH; 0.5 or 1.0 mM) on vegetative growth characteristics of common beans plants grown under different irrigation levels in 2017 (SI) and 2018 (SII) seasons.

| Treatment | Shoot length (cm) | | Number of leaves plant$^{-1}$ | | Leaf area plant$^{-1}$ (dm$^2$) | | Shoot dry weight plant$^{-1}$ (g) | |
|---|---|---|---|---|---|---|---|---|
| | SI | SII | SI | SII | SI | SII | SI | SII |
| $I_{100}$ | 79.3b | 78.3c | 27.0a | 29.3a | 20.7a | 22.5a | 22.8a | 20.4a |
| $I_{80}$ | 78.3b | 76.3cd | 23.7b | 27.0b | 18.3bc | 20.0b | 19.3b | 18.6b |
| $I_{60}$ | 68.3d | 59.7e | 18.3c | 20.0d | 16.3d | 12.9d | 13.2c | 16.6c |
| $I_{100}$ + GSH$_1$ | 86.0a | 87.0a | 27.0a | 28.3a | 21.0a | 23.8a | 23.4a | 21.4a |
| $I_{100}$ + GSH$_2$ | 74.7bc | 80.3bc | 27.3a | 28.0a | 20.6a | 22.9a | 22.2a | 20.0a |
| $I_{80}$ + GSH$_1$ | 78.0b | 78.0c | 27.0a | 29.0a | 20.3a | 22.4a | 22.8a | 18.6b |
| $I_{80}$ + GSH$_2$ | 75.0b | 80.3bc | 26.3a | 29.0a | 20.0ab | 22.6a | 22.0a | 20.4a |
| $I_{60}$ + GSH$_1$ | 70.0cd | 71.3d | 22.3b | 24.0c | 19.5ab | 20.4b | 23.2a | 19.9a |
| $I_{60}$ + GSH$_2$ | 72.0cd | 63.0e | 22.7b | 23.7d | 17.1cd | 15.5c | 19.8b | 16.7c |

Notes.
\# Mean values in each column followed by a different lower-case-letter are significantly different by Tukey's Honest Significant Difference test at $P \leq 0.05$.
$I_{100}$, irrigation with 100% of ETc; $I_{80}$, irrigation with 80% of ETc; $I_{60}$, irrigation with 60% of ETc.

**Table 4** Effect of exogenous spray applications of glutathione (GSH; 0.5 or 1.0 mM) on the productivity of common beans plants grown under different irrigation levels in 2017 (SI) and 2018 (SII) seasons.

| Treatment | Number of pods plant$^{-1}$ | | Green pods weight plant$^{-1}$ | | Green pods yield (ton ha$^{-1}$) | |
|---|---|---|---|---|---|---|
| | SI | SII | SI | SII | SI | SII |
| $I_{100}$ | 26.7$^*$a | 29.3a | 49.0a | 51.7a | 9.61a | 10.15a |
| $I_{80}$ | 19.0b | 22.3b | 40.7b | 41.0b | 8.63b | 9.04b |
| $I_{60}$ | 10.7c | 11.0c | 25.8e | 26.7d | 4.87c | 5.65d |
| $I_{100}$ + GSH$_1$ | 27.0a | 29.7a | 50.7a | 51.1a | 9.73a | 11.02a |
| $I_{100}$ + GSH$_2$ | 28.3a | 29.3a | 49.7a | 50.7a | 9.73a | 10.70a |
| $I_{80}$ + GSH$_1$ | 27.7a | 28.9a | 50.3a | 51.3a | 10.87a | 10.97a |
| $I_{80}$ + GSH$_2$ | 27.3a | 29.0a | 49.7a | 51.0a | 9.93a | 11.00a |
| $I_{60}$ + GSH$_1$ | 19.0b | 22.3b | 36.0c | 36.3c | 8.20b | 8.99b |
| $I_{60}$ + GSH$_2$ | 19.0b | 21.7b | 31.7d | 37.0c | 7.97b | 8.02c |

Notes.
*Mean values in each column followed by a different lower-case-letter are significantly different by Tukey's Honest Significant Difference test at $P \leq 0.05$.
$I_{100}$, irrigation with 100% of ETc; $I_{80}$, irrigation with 80% of ETc; $I_{60}$, irrigation with 60% of ETc.

## Chlorophyll a fluorescence and SPAD value

Data of chlorophyll fluorescence (*i.e.,* Fv/Fm and PI) and SPAD chlorophyll content of common bean plants in response to the application of GSH and water deficits are shown in Table 6. Compared to fully irrigated plants ($I_{100}$), common bean plants subjected to water deficits ($I_{80}$ and $I_{60}$) exhibited lower values of Fv/Fm (by 3% and 7%), PI (24% and 42%), and SPAD value (by 29% and 47%) (seasons average), respectively. However, foliage spraying GSH was observed to adjust the drought-impacted Fv/Fm, PI, and SPAD values of bean plants. In this respect, spraying with 0.5 or 1 mM GSH to water-stressed bean plants at 20% showed similar values of the SPAD value, Fv/Fm, and PI to bean plants subjected to full irrigation without GSH application ($I_{100}$).

**Table 5** Effect of exogenous spray applications of glutathione (GSH; 0.5 or 1.0 mM) on membrane stability index (MSI), relative water content (RWC) and irrigation use efficiency (IUE) of common beans plants grown under different irrigation levels in 2017 (SI) and 2018 (SII) seasons.

| Treatment | MSI (%) | | RWC (%) | | IUE (kg pods m$^{-3}$ of water) | |
|---|---|---|---|---|---|---|
| | SI | SII | SI | SII | SI | SII |
| $I_{100}$ | 52.3[*]a | 57.9a | 87.9a | 89.8a | 2.57d | 3.35de |
| $I_{80}$ | 46.4b | 47.8b | 83.6b | 84.1b | 2.90cd | 3.76c |
| $I_{60}$ | 30.6d | 36.0d | 72.7c | 75.2c | 2.08e | 3.14e |
| $I_{100} + GSH_1$ | 53.7a | 58.6a | 88.3a | 89.7a | 2.78cd | 3.74c |
| $I_{100} + GSH_2$ | 52.8a | 57.4a | 88.5a | 88.9a | 2.78cd | 3.57cd |
| $I_{80} + GSH_1$ | 52.2a | 57.2a | 87.2a | 88.3a | 3.88a | 4.16b |
| $I_{80} + GSH_2$ | 51.9a | 56.7a | 86.1a | 88.0a | 3.55ab | 4.58a |
| $I_{60} + GSH_1$ | 46.3b | 47.5b | 86.9a | 84.1b | 3.43b | 4.44ab |
| $I_{60} + GSH_2$ | 40.7c | 42.7c | 83.1b | 83.9b | 3.13bc | 4.52a |

Notes.

*Mean values in each column followed by a different lower-case-letter are significantly different by Tukey bc Honest Significant deffghgi]g test at $P \leq 0.05$.

$I_{100}$, irrigation with 100% of ETc; $I_{80}$, irrigation with 80% of ETc; $I_{60}$, irrigation with 60% of ETc.

**Table 6** Effect of exogenous spray applications of glutathione (GSH; 0.5 or 1.0 mM) on chlorophyll a fluorescence and relative chlorophyll content (SPAD value) of common beans plants grown under different irrigation levels in 2017 (SI) and 2018 (SII) seasons.

| Treatment | Fv/Fm | | PI | | SPAD chlorophyll | |
|---|---|---|---|---|---|---|
| | SI | SII | SI | SII | SI | SII |
| $I_{100}$ | 0.81[*]a | 0.82a | 2.41b | 2.54a | 35.3a | 35.4a |
| $I_{80}$ | 0.79b | 0.79c | 1.60c | 2.17b | 25.1c | 25.2c |
| $I_{60}$ | 0.76c | 0.76d | 1.15d | 1.72c | 16.8d | 20.5d |
| $I_{100} + GSH_1$ | 0.82a | 0.82a | 2.41b | 2.58a | 37.3a | 36.7a |
| $I_{100} + GSH_2$ | 0.82a | 0.83a | 2.76a | 2.64a | 35.9a | 36.2a |
| $I_{80} + GSH_1$ | 0.81a | 0.82a | 2.42b | 2.51a | 34.1a | 34.7a |
| $I_{80} + GSH_2$ | 0.82a | 0.82a | 2.41b | 2.55a | 34.2a | 34.5a |
| $I_{60} + GSH_1$ | 0.81a | 0.80bc | 1.62c | 2.18b | 31.0b | 29.4b |
| $I_{60} + GSH_2$ | 0.80ab | 0.79c | 1.20d | 2.17b | 31.1b | 28.7b |

Notes.

*Mean values in each column followed by a different lower-case-letter are significantly different by Tukey's Honest Significant Difference test at $P \leq 0.05$.

$I_{100}$, irrigation with 100% of ETc; $I_{80}$, irrigation with 80% of ETc; $I_{60}$, irrigation with 60% of ETc.

## Plant defense system: osmolytes and antioxidants

Figures 1 and 2 show that the contents of total soluble sugars, free proline, AsA, and GSH (not significant) increased in *Phaseolus vulgaris* leaves in response to the water deficit exposure ($I_{80}$ and $I_{60}$), whereas the contents of total free amino acids decreased ($I_{100}$). Under water deficits, exogenous GSH-mediated further increases in total free amino acids, free proline, and total soluble sugars, as well as AsA and GSH compared to stressed plants untreated with GSH ($I_{80}$ and $I_{60}$). In this regard, the maximum values of free proline, AsA, and GSH corresponded with the integrative $I_{60} + GSH_1$ treatment while the integrative $I_{80} + GSH_1$ or $GSH_2$ produced the highest total soluble sugars and total free amino acids levels.

Drought stress ($I_{80}$ and $I_{60}$) increased significantly the activities of antioxidant enzymes in terms of SOD, CAT, APX, and GPX in common bean plants compared to normal

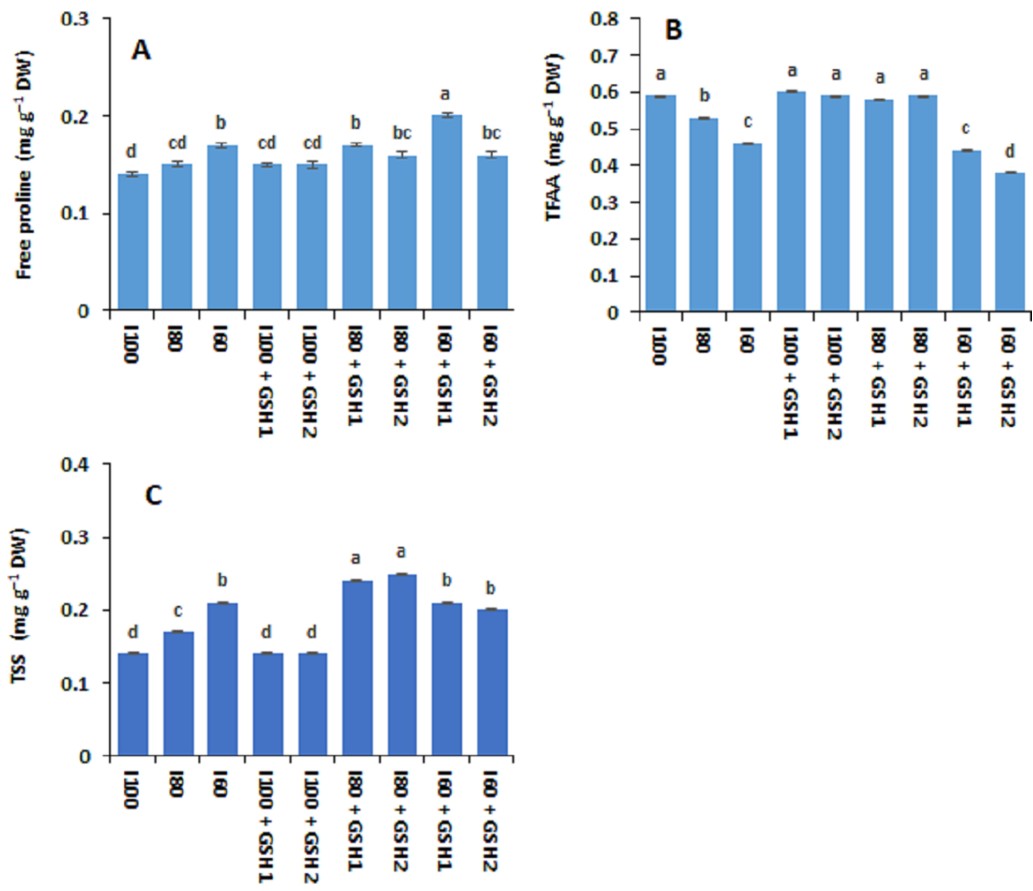

**Figure 1** **Effect of exogenous spray applications of glutathione (GSH; 0.5 or 1.0 mM) on the contents of free proline (A), total free amino acids (TFAA) (B), and total soluble sugars (TSS) (C) of common beans plants grown under different irrigation levels (seasons average).** The vertical bar represents the standard error. Different letters on the bar indicate a significant difference by Tukey's Honest Significant Difference test at $P \leq 0.05$.

conditions ($I_{100}$) as shown in Fig. 3. The aforementioned enzymes activities were at lower level corresponding to the $I_{100}$, $I_{100}$ + $GSH_1$, or $I_{100}$ + $GSH_2$ treatments. Nonetheless, exogenous GSH to water-stressed *Phaseolus vulgaris* induced additional increases in the antioxidative compounds. The magnitude of antioxidant enzyme activity response was more pronounced with integrative $I_{60}$ + $GSH_1$ treatment, followed by integrative $I_{60}$ + $GSH_1$, and $I_{80}$ + $GSH_1$ or $GH_2$ treatments.

## DISCUSSION

As expected, decreasing soil water content under high soil salinity conditions (6.22 dS m$^{-1}$; Table 1) significantly reduced bean plant growth (*i.e.*, shoot height, leaf area, number of leaves, and dry biomass), pods yield, and the IUE (Tables 3–5). Environmental stresses; drought and salinity mediate the loss of cell turgor and impede cell division and elongation, consequently diminishing bean growth and development (*Taiz & Zeiger, 2010*; *Dawood, Abdelhamid & Schmidhalter, 2014*; *Fahad et al., 2017*). However, externally-applied GSH
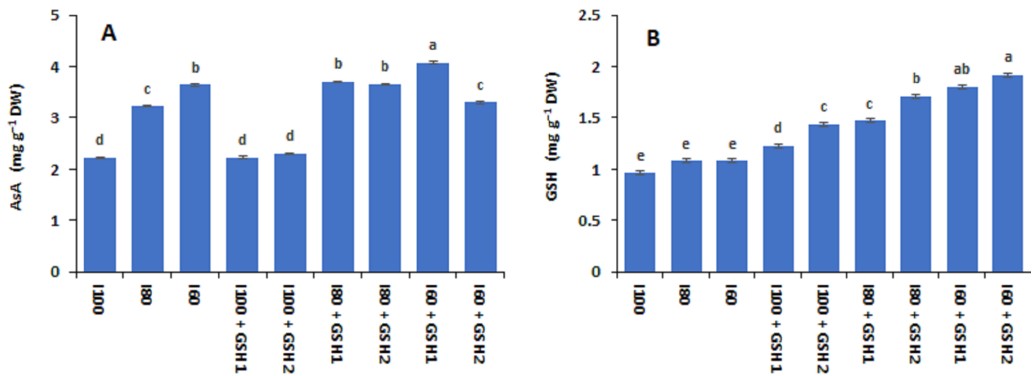

**Figure 2** **Effect of exogenous spray applications of glutathione (GSH; 0.5 or 1.0 mM) on the contents of ascorbic acid (AsA) (A) and glutathione (GSH) (B) of common beans plants grown under different irrigation levels (seasons average).** The vertical bar represents the standard error. Different letters on the bar indicate a significant difference by Tukey's Honest Significant Difference test at $P \leq 0.05$.

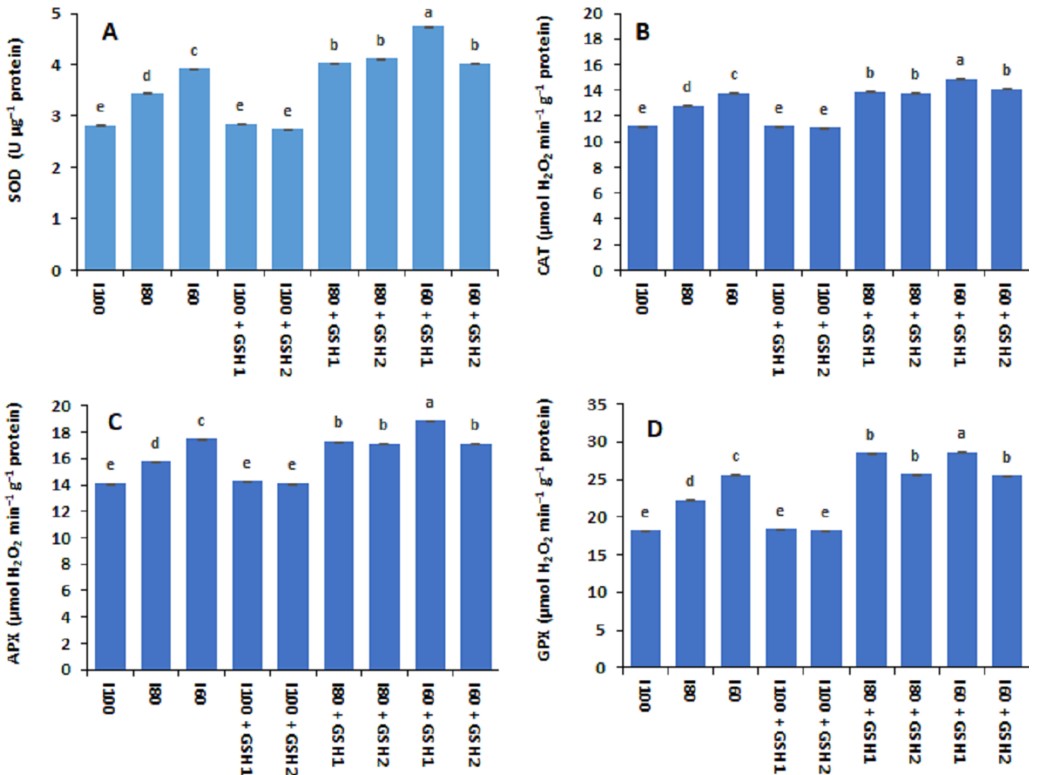

**Figure 3** **Effect of exogenous spray applications of glutathione (GSH; 0.5 or 1.0 mM) on the activities antioxidant enzymes; superoxide dismutase (SOD) (A), catalase (CAT) (B), ascorbate peroxidase (APX) (C), and glutathione peroxidase (GPX) (D) of common beans plan.** The vertical bar represents the standard error. Different letters on the bar indicate a significant difference by Tukey's Honest Significant Difference test at $P \leq 0.05$.

ameliorated drought-induced damage to bean plants, showing that it increased their growth, dry matter, and productivity (yield and IUE) (Tables 3–5) due to increased membrane integrity, tissue water content, and photosynthetic efficiency (Tables 4–6). The application of GSH to water-stressed plants at 20%, yielded better results, whereas when applied to severe water deficit (60% of ETc) improved growth characteristics but not to the same extent as observed under fully irrigated plants. Growth and productivity restoration of water-stressed bean plants by GSH application demonstrated that GSH may involve in mechanisms for drought stress tolerance. GSH is thought to have growth-regulating properties, as increased endogenous GSH promotes cell division in the apical meristem of the root (*Vernoux et al., 2000*; *Hossain et al., 2017*), this root elongation is a key morphological adaptation to water deficit (*Hasanuzzaman, Nahar & Anee, 2017*).

Exogenous GSH increased yield-linked parameters, which could be consequent to the increase of plant leaf area, as well as higher photosynthetic pigments content. Both of these characteristics contribute to increased photosynthetic efficiency and sink capability, which is met by a steady supply of metabolites necessary for the development of bean pods (*Thomas & Howarth, 2000*; *Rehman et al., 2021*). Our results are consistent with a previous study (*Al-Elwany et al., 2020*), which found that foliarly applied GSH increased leaf number, leaf area, shoot dry weight, fruit yield and water use efficiency of chili pepper under water deficits.

Soil water deficit and salinity stress provoke lower cell turgor pressure and soil water potential, increasingly less available water and nutrients acquisition by the plant, thus decreasing the leaf's RWC (Table 5) (*Mahajan & Tuteja, 2005*; *Giménez, Gallardo & Thompson, 2013*; *Sarker & Oba, 2018*). This water stress within the membrane lipid bilayer may result in the degradation of the membrane protein by ROS activity which triggers lipid peroxidation and loss of membrane integrity (Table 5) or even induces complete membrane denaturation (*Mahajan & Tuteja, 2005*; *Abdelkhalik et al., 2020*). However, the negative impacts of decreased RWC and increased cell membranes injured by drought stress were ameliorated in GSH-treated bean plants under water deficit. Therefore, fully irrigated or water-stressed plants at 20% treated with GSH achieved the highest RWC and MSI followed by integrative $I_{60} + GSH_1$ or $GH_2$. High endogenous GSH levels were linked to the regulation of leaf RWC, suggesting that GSH can play a role in leaf rolling control induced by drought stress (*Hasanuzzaman, Nahar & Anee, 2017*). Exogenous GSH has been shown to improve RWC, MSI, and electrolyte leakage in similar studies (*Sohag et al., 2020*; *Rehman et al., 2021*). These improvements in water status and stable cell membrane could be attributed to foliar-applied GSH increased osmolytes accumulation (Fig. 1), which improves osmotic adjustment for maintaining turgor pressure and increases water diffusion into the cell while also increasing antioxidant activities (Figs. 2 and 3) for ROS detoxification and protecting cell membranes (*Nahar et al., 2015*; *Pei et al., 2019*; *Rehman et al., 2021*). These findings highlight the role of GSH in stabilizing membrane integrity for normal functions and increasing tissue water content as metabolically available water, allowing plants to maintain physiological and metabolism activities (*Abid et al., 2018*; *Huang et al., 2020*).

Water deficits reduced the relative chlorophyll content and photosynthetic capacity of the PSII including Fv/Fm and PI. This could be attributed to reduced essential nutrient uptake by roots that are necessary for chlorophyll biosynthesis (*Etienne et al., 2018*; *Semida et al., 2021b*), concomitant with inhibition of D1 protein synthesis and degradation of the thylakoid membrane owing to oxidative stress (*Tian et al., 2013*; *Wang et al., 2018*), indicating inhibition of electron transport chain and the light-harvesting complex of PSII in drought-stressed common bean plants. Nevertheless, GSH-mediated recovery of the relative chlorophyll content and photosynthetic capacity of common bean plants, showed that GSH increased the SPAD chlorophyll, maximal quantity yield (Fv/Fm), and the performance index (PI) of the PSII (Table 6). These findings may be linked to improving plant water status (RWC) and cell membrane stability index (MSI, Table 5) by exogenous GSH for restoring the damaged chloroplast and increasing chlorophyll contents (*Nahar et al., 2015*; *Al-Elwany et al., 2020*). Besides the role of GSH in scavenging ROS *via* the AsA-GSH cycle, GSH-mediated increases in the antioxidative compounds; enzymatic and non-enzymatic (Figs. 2 and 3) that participated in ROS scavenging and inhibition of chlorophyll degradation (*Hasanuzzaman, Nahar & Anee, 2017*; *Hossain et al., 2017*) thus increasing the photosynthetic efficiency. The increment in root growth and biomass has been linked to the up-regulation of cysteine and GSH concentrations to contract water stress in maize plants (*Ahmad et al., 2016*), which may increase water and element uptake and enhance photosynthetic capacity.

In osmotically-stressed common bean plants, clear increases in osmolytes accumulation; total free amino acids, proline, and total soluble sugars were observed in GSH-treated plants. These osmoprotectants might help in water stress tolerance in bean plants by enhancing osmotic adjustment for maintaining cell turgor, protecting cell membranes and proteins against ROS-induced oxidative damage, thus recovering the cellular functions and metabolism as adaptive mechanisms under stressors (*Turner, 2018*; *Sharma et al., 2019*). In previous researches, exogenous GSH was found to up-regulate the accumulation of osmolytes such as proline in mung beans grown under high-temperature stress (*Nahar et al., 2015*), proline and soluble sugars in chili pepper grown under water stress (*Al-Elwany et al., 2020*). Also, higher osmolytes concentration after foliar spraying GSH may have a valuable role in ameliorating oxidative damage and acting as an osmoprotectant to prevent water loss and increase the RWC (*Nahar et al., 2015*; *Turner, 2018*).

Antioxidant defense system components help to lessen oxidative damage and confer drought stress tolerance in plants. Among them, GSH is one of the most remarkable antioxidants, which forms a crucial portion of the AsA-GSH cycle (*Bartoli et al., 2017*; *Hasanuzzaman, Nahar & Anee, 2017*). Common bean plants exposed to water deficits exhibited higher GSH and AsA contents than those well-watered. However, exogenous GSH application mediated further elevation in AsA and GSH content in drought-stressed plants (especially at severe levels), indicating an enhancement in the AsA-GSH cycle, and up-regulation associated enzymes activity as an effective pathway to ameliorate the oxidative damage in cellular organelles under abiotic stress (*Hasanuzzaman et al., 2019*; *Semida et al., 2021a*). GSH and AsA predominately quench $O_2$ directly or by enzyme catalysis (*Sarker & Oba, 2018*). GSH and AsA have high redox potentials, and interactions with a variety

of components and pathways to maintain a normally reduced state. As a result, AsA and GSH correlate with the activity of various enzymes (DHAR, GR and MDHAR, and APX) during the ROS detoxification process by donating electrons or reducing equivalents (*Foyer & Noctor, 2011*; *Hasanuzzaman et al., 2019*). Supplementation of GSH to water-stressed mung bean markedly decreased the indicators of oxidative stress; $H_2O_2$, OH −, $O2^{\bullet--}$, and lipid peroxidation, showing the effective role of GSH in reducing oxidative damage (*Nahar et al., 2015*).

Furthermore, our results exhibited that foliar-applied GSH elevated the enzymatic antioxidant; SOD, CAT, APX, and GPX capacity in leaves of *Phaseolus vulgaris* plants grown under water deficit (Fig. 3), which is paramount in the removal of ROS in plant tissues. The balance among SOD and APX or CAT activities in the plant cell is critical to determine the stable state of the level of $H_2O_2$ and $O2^{\bullet--}$ to inhibit the forming of the highly toxic $OH^-$ (*Mittler, 2002*). The SOD is found in nearly all tissues and serves as the primary line of defense in the ROS detoxification approach by disputing the $O2^{\bullet--}$ into $O_2$ and $H_2O_2$. After that, the CAT dismutase the $H_2O_2$ to $H_2O$ during stress, or the $H_2O_2$ enters the AsA-GSH cycle, where the APX utilizes the AsA as an electron donor to convert $H_2O_2$ to $H_2O$ (*Das & Roychoudhury, 2014*; *Hasanuzzaman, Nahar & Anee, 2017*). Also, GPX used the GSH as a substrate during scavenging $H_2O_2$ and lipid hydroperoxides (*Noctor et al., 2012*; *Zhang et al., 2019*), thus increasing the endogenous GSH associated with up-regulation of the activity of GPX may help to scavenge ROS in bean plants under water stress.

In the present study, it was observed that the higher concentration of glutathione led to similar or lower values than the lower concentration. The response of plants to different GSH concentrations varies depending on the crop (*Hasanuzzaman, Nahar & Anee, 2017*). In stress responses to GSH, an initial response phase (change in GSH redox state) will be followed by an acclimatization phase in which a new steady state is established (increase GSH level and related enzymes activity or/and more reduced GSH redox state). Alternately, system degradation will occur if successful acclimatization is not accomplished (*Tausz, Šircelj & Grill, 2004*). According to various studies, GSH levels may increase, not change, or decrease under stress. GSH redox potential, which depends on both the GSH/GSSG ratio and GSH concentration, and the redox state of GSH/GSSG may change to become more oxidized, more reduced, or not change at all (*Dorion, Ouellet & Rivoal, 2021*; *Ito & Ohkama-Ohtsu, 2023*). Indeed, further investigations with more related measurements are needed to determine the exact reasons for the difference of some parameters under different GSH concentrations. Externally-applied GSH alleviated the water stress (especially at a moderate level) on bean plants grown under salt-affected soil conditions, while under acute water stress improved to some extent the growth and productivity of bean plants.

## CONCLUSIONS

In summary, reducing irrigation to common bean plants under moderate soil salinity conditions decreased the leaf water content, membrane integrity, SPAD chlorophyll, and photosynthetic efficiency of the PSII, resulting in reduced growth and pods yield while

not improving the irrigation use efficiency. On the other hand, exogenously applied GSH mitigated the adverse effects of deficit irrigation on common beans, GSH-directed improvements in the growth, yield, and physio-biochemical properties of bean plants. The relative water content, membrane stability, and photosynthetic efficiency of the PSII were increased in water-stressed bean plants by GSH application. GSH-induced drought stresses tolerance by up-regulating total free amino acids, free proline, and total soluble sugars, as well as AsA and GSH and enzymatic antioxidants (APX, CAT, SOD, and GPX) for osmotic adjustment and stabilizing membrane integrity of bean plants. The integrative $I_{80}$ + $GSH_1$ or $GH_2$ were more pronounced, recording similar or higher values than well-watered plants untreated with GSH($I_{100}$). Therefore, integrating GSH and water deficit is recommended for future application in order to improve *Phaseolus vulgaris* performance under soil salinity conditions.

### Funding
This study was a portion of Research Project No. 11030129 corroborative by the National Research Centre, Cairo, Egypt. The funders had no role in study design, data collection and analysis, decision to publish, or preparation of the manuscript.

### Grant Disclosures
The following grant information was disclosed by the authors:
National Research Centre, Cairo, Egypt: 11030129.

### Competing Interests
Magdi Abdelhamid & Othmane Merah are Academic Editors for PeerJ.

### Author Contributions
- Taia A. Abd El Mageed conceived and designed the experiments, performed the experiments, prepared figures and/or tables, and approved the final draft.
- Wael Semida conceived and designed the experiments, performed the experiments, prepared figures and/or tables, and approved the final draft.
- Khoulood A. Hemida conceived and designed the experiments, performed the experiments, prepared figures and/or tables, and approved the final draft.
- Mohammed A.H. Gyushi conceived and designed the experiments, performed the experiments, prepared figures and/or tables, and approved the final draft.
- Mostafa M. Rady conceived and designed the experiments, performed the experiments, prepared figures and/or tables, and approved the final draft.
- Abdelsattar Abdelkhalik conceived and designed the experiments, performed the experiments, prepared figures and/or tables, and approved the final draft.
- Othmane Merah analyzed the data, authored or reviewed drafts of the article, and approved the final draft.
- Marian Brestic analyzed the data, authored or reviewed drafts of the article, and approved the final draft.

- Heba I. Mohamed analyzed the data, authored or reviewed drafts of the article, and approved the final draft.
- Ayman El Sabagh analyzed the data, authored or reviewed drafts of the article, and approved the final draft.
- Magdi T. Abdelhamid conceived and designed the experiments, performed the experiments, analyzed the data, prepared figures and/or tables, authored or reviewed drafts of the article, and approved the final draft.

## Data Availability

The raw data are available in the Supplemental File.

## Supplemental Information

Supplemental information for this article can be found online at http://dx.doi.org/10.7717/peerj.15343#supplemental-information.

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
