# Peer review of "Glutathione-mediated changes in productivity, photosynthetic efficiency, osmolytes, and antioxidant capacity of common beans (Phaseolus vulgaris) grown under water deficit"

_PeerJ, doi:10.7717/peerj.15343_

## Round 0.1 · original submission · Major Revisions

Although the reviewers have recommended reconsideration of the paper, the paper will only be considered if the language of the manuscript is improved. Please take help from an expert.

·

Basic reporting

The manuscript entitled “Glutathione-mediated changes in productivity, photosynthetic efficiency, osmolytes, and antioxidant capacity of common beans (Phaseolus vulgaris) grown under water deficit ” seems to be very interesting research article in the field of agriculture, horticulture and all allied field. The manuscript is compiled technically with comprehensive information supported with up to-date literature. The manuscript is comprised of 3 sections: Introduction; Material and Method, and Results and Discussion. Introduction contain well compiled information indicating that authors are well aware about the study conducted. Material and method contain information about experimental design and methodology adopted. It may attract the attentions of readers, researchers, and other workers in the relevant fields. It needs minor revisions before publication in this journal because journal is committed to publish the well oriented publications.

Experimental design

Material and method contain information about experimental design and methodology adopted.

2. How crop was identified? There are many varieties of Common Beans. In the present study, which variety was selected??
4. Materials and Methods should be written as “Material and Methods”.
5. Methodology seems to be more complex. Authors should write in easily understandable way. It will attract the readers in future.
6. Brief Methodology in the Abstract should be added.

Validity of the findings

The manuscript is compiled technically with comprehensive information supported with up to-date literature. The manuscript is comprised of 3 sections: Introduction; Material and Method, and Results and Discussion. Introduction contain well compiled information indicating that authors are well aware about the study conducted.

Additional comments

1. Language of manuscript needs improvement.
2. How crop was identified? There are many varieties of Common Beans. In the present study, which variety was selected??
3. In the Abstract, future insights should be clearly added. What are future perspectives?
4. Materials and Methods should be written as “Material and Methods”.
5. Methodology seems to be more complex. Authors should write in easily understandable way. It will attract the readers in future.
6. Brief Methodology in the Abstract should be added.
7. Discussion may be improved by comparing and discussing with more recent studies conducted on Common beans.
8. References should be cross checked and verified according to Journal format.
9. Conclusion need improvement. What was concluded in the present work? Future aspects and recommendations should be added.

Reviewer 2 ·

Basic reporting

The manuscript by Taia Abd El-Mageed et.al. reported that the exogenous application of Glutathione to common beans resulted in improved drought tolerance and total yield. The study is important for optimizing agricultural productivity, as well as understanding the mechanism of drought/salt stress responses.

There are a few minor comments listed below. Hope the authors could address before considered for publication:
1. The writing should be improved to ensure clear understanding. Several examples where the sentences were difficult to understand can be found in line 108-109, line 199-200, and line315-316.
2. I would also suggest the authors modify the Abstract to a more succinct format so audiences could catch the main points.
3. Please provide more details on the application of GSH. The authors only mentioned that the selected concentrations performed well in the preliminary experiment (not shown), but did not elaborate the method in current study (I.e., was GSH dissolved in water or other solvent for spray? What was the volume of GSH solution used for each plant/treatment?)
4. The higher GSH concentration (GSH2) did not show better performance comparing to lower ones (GSH1) in the same irrigation condition, and it was even worse in some cases (i.e. Table 5). Could the authors suggest any explanation for the absence of positive association between GSH concentration and drought tolerance?

Experimental design

no comment

Validity of the findings

no comment

---

## Round 0.2 · accepted · Accept

Authors have revised the manuscript very well. Therefore, I recommend that the manuscript can be accepted for publication in PeerJ.

·

Basic reporting

Dear Editor
I have gone through the revised manuscript, submitted by the authors. The revised manuscript showed that it has been revised in the light of queries by the reviewer. All the queries raised are answered properly. I recommended the research article for publication in the journal after the completion of other codal formalities

Experimental design

Satisfactory

Validity of the findings

Original and revised according to queries

Additional comments

I have gone through the revised manuscript, submitted by the authors. The revised manuscript showed that it has been revised in the light of queries by the reviewer. All the queries raised are answered properly. I recommended the research article for publication in the journal after the completion of other codal formalities

Reviewer 2 ·

Basic reporting

The authors have revised the writing and added more detailed description of experimental design and methods. All the comments have been addressed well.

Suggestions for future experiments:
1. Since the GSH was dissolved in 9L distilled water for every 9 square meter plot, I would suggest the authors to include one control treatment (with 9L water+Tween-20, but no GSH) in future experiments, in order to confirm that the improved drought responses are truly from GSH treatment rather than those additional 9L water.
2. The higher GSH concentration showed reduced effect, therefore it might be helpful to test a series of dilution to find out the optimal GSH concentration in future experiments.

Experimental design

no comments

Validity of the findings

no comments